

# Reversal of ocean gyres near ice shelves in the Amundsen Sea caused by the interaction of sea ice and wind

Yixi Zheng[1], David P. Stevens[2], Karen J. Heywood[1], Benjamin G. M. Webber[1], Bastien Y. Queste[3]

[1] Centre for Ocean and Atmospheric Sciences, School of Environmental Sciences, University of East Anglia, Norwich NR4 7TJ, UK
[2] Centre for Ocean and Atmospheric Sciences, School of Mathematics, University of East Anglia, Norwich NR4 7TJ, UK
[3] Department of Marine Sciences, University of Gothenburg, Box 460, 405 30 Göteborg, Sweden.

*Correspondence to*: Yixi Zheng (Yixi.Zheng@uea.ac.uk)

**Abstract**. Floating ice shelves buttress the Antarctic Ice Sheet, which is losing mass rapidly mainly due to ocean-driven melting and the associated disruption to glacial dynamics. The local ocean circulation near ice shelves is therefore important for the prediction of future ice mass loss and related sea-level rise as it determines the water mass exchange, heat transport under the ice shelf and the resultant melting. However, the dynamics controlling the near-coastal circulation are not fully understood. A cyclonic (i.e. clockwise) gyre circulation (27 km radius) in front of the Pine Island Ice Shelf has previously been identified in both numerical models and velocity observations. Here we present ship-based observations from 2019 to the west of Thwaites Ice Shelf, revealing another gyre (13 km radius) for the first time in this habitually ice-covered region, rotating in the opposite (anticyclonic, anticlockwise) direction to the gyre near Pine Island Ice Shelf, despite similar wind forcing. We use an idealised configuration of MITgcm, with idealised forcing based on ERA-5 climatological wind fields and simplified sea ice conditions from MODIS satellite images, to reproduce key features of the observed gyres near Pine Island Ice Shelf and Thwaites Ice Shelf. The model driven solely by wind forcing in the presence of ice can reproduce the horizontal structure and direction of both gyres. We show that the modelled gyre direction depends upon the spatial difference in the ocean surface stress, which can be affected by the applied wind stress curl filed, the percentage of wind stress transferred through the ice, and the angle between the wind direction and the sea ice edge. The presence of ice, either it is fast ice/ice shelves blocking the effect of wind, or the mobile sea ice enhancing the effect of wind, has the potential to reverse the gyre direction relative to ice-free conditions.

## 1 Introduction

Antarctic ice shelves are thinning rapidly due primarily to basal melting, allowing the ice sheets to accelerate and lose mass (e.g. Pritchard et al., 2012) to significantly contribute to the future sea-level rise (e.g. Bamber et al., 2019; Golledge et al., 2019; DeConto et al., 2021). The highest thinning rate has been observed among those ice shelves draining towards the Amundsen Sea (e.g. Rignot et al., 2019; Paolo, Fricker and Padman, 2015), where relatively warm modified Circumpolar Deep Water (mCDW) intrudes onto the Amundsen Sea continental shelf via bathymetric troughs, allowing it to come into direct





contact with the base of ice shelves (e.g. Rignot et al., 2019; Heywood et al., 2016). The flux of the modified Circumpolar Deep Water (mCDW) entering the ice shelf cavity determines the rate of ice shelf melting (e.g. Jacobs et al., 2011, Dutrieux et al., 2014). Therefore, understanding the local circulation that determines the flow of mCDW and its associated heat transport toward the ice shelves is crucial for a better prediction of ice shelf melting, future sea level and climate.

Models and observations have revealed the presence of a cyclonic gyre in the centre of Pine Island Bay (PIB; hereafter PIB gyre for this gyre). The gyre is well defined between the sea surface and about 700-m depth, in front of Pine Island Ice Shelf (Thurnherr et al., 2014; Heywood et al., 2016). Gyres play an important role in local ocean circulation, distributing heat and enhancing water mass exchange in the Amundsen Sea (Zheng et al., 2021; Schodlok 2012). Schodlok et al. (2012) use a high-resolution model to infer that the strength of the PIB gyre is the main determinant of heat transport toward the ice shelf

and the associated glacial melt rate (Schodlok et al., 2012). Zheng et al. (2021), Mankoff et al. (2012) and Tortell et al. (2012) also suggest that the PIB gyre entrains water as it exits the ice cavity contributing to the spreading of glacial meltwater and its associated heat, nutrients and freshwater.

    The formation of the PIB gyre has been attributed to the wind forcing and the meltwater outflow in the south-eastern Amundsen Sea (e.g. Thurnherr et al., 2014). Model results from Heimbach and Losch (2012) show that the PIB gyre would be

weaker by more than two-thirds if wind forcing were absent. From Oct 2011 to May 2013, moored current meters in PIB revealed a reversal of the ocean current velocity (Webber et al., 2017), potentially indicating a change of PIB gyre direction from cyclonic to anticyclonic. However, the driver of any reversal is still uncertain as the limited available observations do not show a change in the sign of the wind stress curl field nor in meltwater outflow locations.

    The sparse spatial coverage of observations in these often-ice-covered regions limits our understanding of the mechanisms

regulating these gyres. In Sect. 2, we present a set of new observations of another gyre near of Thwaites Ice Shelf (hereafter Thwaites gyre; Fig. 1). Influenced by the same climatologically-cyclonic wind field, Thwaites gyre is anticyclonic, raising the intriguing question of what mechanism(s) control(s) the direction of these gyres. We note the different sea ice coverage over the Thwaites and PIB gyres, and hence hypothesise that sea ice can influence the wind stress field felt by the ocean (i.e. ocean surface stress, hereafter OSS) to alter the gyre direction. Considering the sparse observations in the Amundsen Sea, we adopt

a new approach to this complex question. In Sect. 3, we introduce an idealised model designed to explore the roles of wind and sea ice in determining the gyre features. Sect. 4 presents the results of the idealised model when different ice conditions and wind stress fields are applied. We discuss the limitations and applications of the results and summarise the results in Sect. 5.

## 2 Observations of gyres in the west of Thwaites Ice Tongue

From 25 February to 4 March 2019, the RV Nathaniel B. Palmer collected the first hydrographic dataset to the west of Thwaites Ice Tongue as part of the International Thwaites Glacier Collaboration: Thwaites-Amundsen Regional Survey and Network (ITGC: TARSAN) project. Temperature and salinity profiles were obtained using a Sea-Bird SBE 911+ CTD (Conductivity



Temperature Depth) profiler with two pairs of conductivity and temperature sensors and then vertically averaged into 1-dbar bins. Ocean current velocities over the upper 980 m of the water column were obtained using a 75 kHz Ocean Surveyor (OS75) and a 38 kHz Ocean Surveyor (OS38) shipboard Acoustic Doppler Current Profiler (sADCP) and then horizontally averaged into 2 km × 2 km bins. No de-tiding has been applied to the velocity measurements presented in this study because the moored current meters and models in the region suggest that tidal currents are less than 2 cm s$^{-1}$ (Jourdain et al., 2019) and bathymetry has been so poorly known in this region that uncertainties in predicted tidal model currents are of a similar magnitude.

This region is habitually covered by sea ice and has only opened twice since 2000 (WorldView Aqua/MODIS corrected reflectance). The newly-obtained velocity dataset reveals the previously unreported Thwaites gyre (blue arrows in Fig. 2). Based on the observations, we identify the centre of Thwaites gyre at -107.55 ºW, -75 ºS. We then calculate the tangential components of ocean current around concentric circles centred on Thwaites gyre and average them into 1-km-radius bins (Fig. 3). Thwaites gyre has an approximate radius of 13 km and can be well identified in the 30 m to 430 m depth range covered by the sADCP (Fig. 3), recirculating about 0.2 Sv (i.e. $10^6 \ m^3 \ s^{-1}$) of water, with the highest speed of about 10 cm s$^{-1}$ (Fig. 3). Note that Thwaites gyre is anticyclonic, despite the local cyclonic wind stress curl, evident in both the climatology (Fig. 1) and contemporaneous observations (Fig. 4), favouring cyclonic gyres.

To identify the role of the gyre in transporting meltwater, and test if meltwater outflow can help to explain the gyre rotation, we calculate meltwater content from temperature and salinity profiles in the gyre region using the composite-tracer method (Jenkins, 1999; Pink dots in Fig. 2). In this calculation, we use three water masses including mCDW, Winter Water and glacial meltwater, and the two tracers: conservative temperature ($\Theta$) and absolute salinity ($S_A$), defined following the Thermodynamic Equations of Seawater-10 standard (McDougall and Barker, 2011). Both tracers are assumed to be conservative for all observations. We chose the endpoints following previously published research. The endpoints of mCDW ($\Theta = 1.044$°C and $S_A = 34.8795$) are consistent with Wåhlin et al. (2021), and the endpoints of Winter Water ($\Theta = -1.86$°C, $S_A = 34.32$) and glacial meltwater ($\Theta = -90.8$°C, $S_A = 0$) are the same as used by Zheng et al. (2021) and Biddle, Loose, and Heywood (2019). The fraction of meltwater can be derived from observations with the equation below:

$$\varphi_{\text{meltwater}} = \frac{\Theta_{\text{observed}} - \Theta_{\text{mCDW}} - \dfrac{\left(S_{A_{\text{observed}}} - S_{A_{\text{mCDW}}}\right) \times \left(\Theta_{\text{WW}} - \Theta_{\text{mCDW}}\right)}{\left(S_{A_{\text{WW}}} - S_{A_{m\text{CDW}}}\right)}}{\Theta_{\text{meltwater}} - \Theta_{\text{mCDW}} - \dfrac{\left(S_{A_{\text{meltwater}}} - S_{A_{m\text{CDW}}}\right) \times \left(\Theta_{\text{WW}} - \Theta_{\text{mCDW}}\right)}{\left(S_{A_{\text{WW}}} - S_{A_{\text{mCDW}}}\right)}} \quad (1)$$

where $\varphi_{\text{meltwater}}$ is the meltwater fraction, and $\Theta$ and $S_A$ with subscripts define the conservative temperature and absolute salinity endpoints of each water mass.

The highest meltwater content is detected in the southeast of the Thwaites gyre (Fig. 2). This is consistent with observations collected by autonomous underwater vehicle presented in Wåhlin et al. (2021) that suggest a north-westward meltwater-rich outflow emanating from the cavity beneath Thwaites Ice Tongue. The Thwaites gyre may entrain this meltwater plume and thus play a role in circulating meltwater near Thwaites Ice Shelf and boost water-mass mixing.



Although a previous study suggests that the buoyancy of glacial meltwater at depth may facilitate gyres (Mathiot et al., 2017), Thwaites gyre is not likely to be meltwater-driven. As glacial meltwater plumes coming out from the base of glacier are more buoyant than the ambient water, they rise and turn left due to Coriolis force, as seen in Pine Island Bay (e.g. Thurnherr et al., 2014, Zheng et al. 2021). Glacial meltwater coming out from Thwaites Ice Tongue to the southwest of the Thwaites gyre will therefore impede the anticyclonic gyre, rather than accelerate it. Hence, neither the cyclonic wind stress curl shown in Fig. 2 nor the meltwater discharge can directly generate an anticyclonic gyre. Therefore, we explore other factors that might explain this apparent contradiction.

Sea ice coverage is often remarkably different between the PIB and Thwaites gyre regions (Fig. 1). Satellite imagery shows that the PIB gyre region was generally open during the whole summer of 2009 (Worldview Aqua/MODIS corrected reflectance), when the PIB gyre was firstly observed (Thurnherr et al., 2014). At the same time, fast ice covered most of the Thwaites gyre region and the sea ice did not open until late January 2019 (Fig. 4a), about a month before the sADCP survey revealed the gyre (in late February to early March 2019). During February, ice coverage in the Thwaites gyre region changed from covering the western part of the Thwaites gyre (01 Feb 2019, Fig. 4b), to completely open (12 Feb 2019, Fig. 4c). Sea ice covered the western part of the Thwaites gyre again (23 Feb 2019, Fig. 4d) two days before the start of the sADCP data collection.

The presence of sea ice alters OSS (e.g. Elvidge et al., 2016; Meneghello et al., 2018). Thus, we hypothesise that the sea ice coverage may mediate the OSS and the resulting surface stress curl felt by the ocean (i.e. ocean surface stress curl, hereafter OSSC) sufficiently to reverse a gyre, leading to the different PIB and Thwaites gyre directions. To test this hypothesis, we use an idealised model to reproduce wind-driven gyres and run a set of conceptual experiments to simulate the response of wind-driven gyres to different sea ice coverages.

# 3 Model experimental design

## 3.1 Model set up

We employ the MIT general circulation model (MITgcm; Marshall et al., 1997) with an idealised barotropic set up. The model has an ocean domain with a size of $60 \times 60$ km and a horizontal grid spacing of 1 km (Fig. 5, for comparison, the baroclinic Rossby radius in this region is about 5 km, following the calculation described by Chelton et al., 1998). It has one 1-km-thick vertical layer with a free surface. The size of the model domain is comparable to the PIB gyre region. The bottom boundary is free-slip with no drag and the lateral boundaries are no-slip. We set the Coriolis parameter $f$ to be $-1.4083 \times 10^{-4} \, s^{-1}$, appropriate for 75°S, with a meridional gradient $\beta \, of \, 1 \times 10^{-11} \, s^{-1} m^{-1}$. The timestep is $120 \, s$. The southern boundary is envisaged to be the ice shelf (Fig. 5).

We run all simulations for six model months, which allows all of them to spin up to be sufficiently close to a steady state. The spin-up time of the simulations varies from 51 days to 91 days, assessed as the time at which the daily change of the total



kinetic energy of the ocean is less than 0.1% of the total kinetic energy of the ocean at the final model day of the 6 model
125    months.

## 3.2 Wind forcing

The wind field is the only external forcing applied to the model ocean. We generate a simplified wind forcing field (Fig. 6a) based on the key features of the climatological wind in the south-eastern Amundsen Sea to include the ice conditions for both Pine Island Bay and around the Thwaites Ice Tongue (Fig. 1). The ERA5 climatological 10-m wind above the PIB and
Thwaites gyres blows from the ice shelves to the ocean, with a speed decreasing from the southeast to the northwest. As mentioned in Sect. 3.1, we rotate the domain relative to true north such that these offshore winds are purely meridional in the model, with zero zonal wind (Fig. 5). The maximum wind speed ($10 \text{ m s}^{-1}$) occurs in the southwestern corner of the model domain (Fig. 6a). The meridional gradient of wind speed ($-1.667 \times 10^{-6} \text{ s}^{-1}$) is one-fifth of the zonal gradient of wind speed ($-8.333 \times 10^{-6} \text{ s}^{-1}$). The meridional wind stress is given by,

$$\tau_y = C_D \rho_{air} |v| v \qquad (2)$$

where $C_D = 1 \times 10^{-3}$ is the drag coefficient, $\rho_{air} = 1.275 \ kg \ m^{-3}$ is the air density and $v$ is the wind speed. The wind forcing field applied in our study can be downloaded from the link in the Data Availability section.

We vary the strength and sign of the wind stress curl to generate four wind forcing fields: strong or weak, cyclonic or anticyclonic wind stress curl (Fig. 6a-d). The simplified wind field representing the climatological conditions in the
southeastern Amundsen Sea is shown in Fig. 6a. The same strength of wind stress curl, but anticyclonic, is shown in Fig. 6c. The two remaining wind fields have wind stress curls weaker by 50% (Fig 6b,d). The average wind speed over the whole ocean model domain is kept the same for all four wind fields.

## 3.3 Sea ice coverage

We do not include an active sea ice model, but only consider the influence of ice on the OSS and OSSC. Fast ice and ice
shelves are unable to move significantly so are expected to completely block the wind stress but other types of ice coverage may have different impacts on the OSS. Previous research has found a generally higher momentum transfer over ice-covered regions than open water due to ice drift dragging the ocean (e.g. Martin, Steele and Zhang, 2014; Meneghello et al., 2018). The magnitude of this additional stress on the ocean surface from ice drift may change due to the different types and concentrations of ice coverage. We vary the magnitude of OSS in the ice-covered half of the model domain from 0% to 200%
of wind stress to sample a range of possible OSS modification by ice in steps of 20%. For all simulations, OSS remains unaltered in the ice-free half.

In addition, we vary the angle between the sea ice edge and wind direction from $\pi/4$ to $3\pi/4$ (Fig. 7a) to generate five different ice coverages as shown in Fig. 7b-f. For all ice coverages applied in this study, the ice covers exactly half the model domain with the sea ice edges always intersecting the centre of the domain.



## 4 Model results

### 4.1 Effect of wind stress on simulated gyres

We first consider results from simulations with no ice coverage. When sea ice is absent, the OSS and OSSC remain unchanged over the whole model domain. The strength and direction of gyres solely depend on the wind field. Cyclonic wind stress curl fields (Fig. 6a,b) generate cyclonic gyres (Fig. 8a,b), stronger wind stress curl fields (Fig. 6a,c) generate stronger gyres (Fig. 8a,c), and vice versa, as expected.

For both *StrongCyclonic* and *StrongAnticyclonic* wind fields, the simulated gyres have a maximum streamfunction of $0.58$ Sv and a maximum current speed of $3.2$ cm s$^{-1}$. Compared with the PIB gyre surveyed in 2009 ($1.5$ Sv, $30$ cm s$^{-1}$, Thurnherr et al., 2014), which is also located in an ice-free area, the simulated gyres are weaker with a slower current speed. The different strength between simulated gyres and PIB gyre might be due to the lack surface intensification of the currents and the lack of meltwater injection in our barotropic model. Nevertheless, our idealised model captures the characteristics of the gyre sufficiently to be a useful tool to explore the effects of different forcing fields and sea ice coverage on gyre strength, shape and direction.

### 4.2 Effect of sea ice on simulated gyres

#### 4.2.1 An example of a simulated gyre similar to the Thwaites gyre

As an example of the sea ice influencing OSSC, we first discuss the simulation that generates an anticyclonic gyre similar to the observations of the Thwaites gyre discussed in Sect. 2. As mentioned in Sect. 2, in early March 2019, sea ice covered the western part of the Thwaites gyre (Fig. 2), at an angle to the wind stress similar to the $SeaIce\frac{\pi}{4}$ ice coverage (Fig. 7f). The ERA5 wind stress curl was about $-5$ to $-7.5 \times 10^{-7}$ N m$^{-3}$, similar to the *WeakCyclonic* wind field (Fig. 6d). We therefore consider the *WeakCyclonic* wind field and the $SeaIce\frac{\pi}{4}$ ice coverage to mimic (in an idealised way) the response of the ocean to the wind and ice conditions in the Thwaites gyre region in March 2019 (Fig. 9).

The OSSC is zero over the ice-covered domain (northwestern half of Fig. 9a) and negative (i.e., cyclonic in southern hemisphere) over the ice-free domain (southeastern half of Fig. 9a). Due to the different OSS between ice-covered and ice-free domains, positive OSSC (i.e., anticyclonic in southern hemisphere) occurs along the sea ice edge, with a magnitude about ten times larger than the negative values occurring in the ice-free domain (Fig. 9a). The magnitude of the OSSC along the sea ice edge decreases from the southwest to the northeast, due to the negative meridional and zonal gradients in wind stress (Fig. 9a).

This asymmetric positive OSSC along the sea ice edge then results in an asymmetric anticyclonic gyre with its centre located slightly left of the centre of the model domain (Fig. 9b). The anticyclonic gyre has a maximum $\Psi$ of $-0.56$ Sv and a maximum current speed of $3.0$ cm s$^{-1}$ at steady state. Thwaites gyre in reality has a higher maximum speed ($10$ cm s$^{-1}$, Fig.



3) but circulates less water (0.2 Sv) than the simulated gyre. This is partly because the thickness of the water column influenced by the Thwaites gyre is only about 400 m, while it is 1000 m in our simulation presented here. We tested the same model set up with a 400-m depth and compare the results with the 1000-m-depth model results. The simulated gyre from 400-m-deep model set up has a faster speed, but the streamfunction and gyre features remain very similar to gyre from 1000-m-depth simulation. Overall, this experiment demonstrates that, even with an idealised barotropic model, the presence of sea ice can

enable a cyclonic wind field to generate an anticyclonic gyre, similar to that observed in 2019.

### 4.2.2    Overview of the effect of sea ice coverage on simulated gyres

The example discussed above illustrates the influence of a single configuration of ice coverage ($SeaIce\frac{\pi}{4}$ and $0\%\tau$) on the simulated gyre. To make our results more generally applicable, we run our model while varying the ice edge angle and the percentage of wind stress transferred through the sea ice. Fig. 10 and Fig. 11 illustrate that wind-driven gyres can reverse

despite an unchanged wind field for a range of parameters. Here we use Fig. 10 and Fig. 11 to present an overview of the simulated gyre features being affected by both the angle between the wind and sea ice edge, and the amount of stress transferred to the ocean. In the following Sect. 4.2.3 and 4.2.4, we discuss the mechanisms underlying these processes in more detail.

We use the ocean model streamfunction to diagnose the strength and direction of the simulated gyre – the maximum magnitude of streamfunction reflects the gyre strength and its sign reflects gyre direction. Some of our simulations generate

two or three connected gyres with different strengths and directions (i.e. dipoles or tripoles). In the analysis we discuss only the gyre with the greatest magnitude of streamfunction (i.e. dominant gyre) in such simulations, unless otherwise stated.

It is well-established that the strength of gyres is closely related to the OSSC integrated over the wind-influenced area (Stommel, 1948). In agreement with this, we find that negative area-integrated OSSCs tend to generate cyclonic gyres while positive area-integrated OSSCs tend to generate anticyclonic gyres (Fig. 10). The simulated maximum streamfunction and the

205 area-integrated OSSC are approximately linearly correlated (Fig. 10), such that stronger OSSC leads to stronger gyres. Although this relationship is very strong, demonstrating the dominant influence of the magnitude of the area-integrated OSSC, the location and distribution of the OSSC does have a secondary influence that accounts for deviations from a perfect correlation.

Figure 11 shows that strong wind stress curl fields (i.e. *StrongCyclonic*, thick lines) lead to stronger gyre strength (i.e.

higher magnitude of streamfunction) than those from weak wind stress curl fields (i.e. *WeakCyclonic*, thin lines). Note that Fig. 11 only contains the simulations with cyclonic wind fields (i.e. *StrongCyclonic* and *WeakCyclonic*). Results from simulations with anticyclonic wind fields (i.e. *StrongAnticyclonic* and *WeakAnticyclonic*) are mirror images of those from simulations with cyclonic wind fields about the 0-Sv line.

Fig. 10 illustrates how the change in area-integrated OSSC depends on the angle between the wind direction and the ice

edge. Because the imposed zonal gradient of wind stress is greater than the meridional gradient in our experiments, the difference between the OSS of ice-covered and ice-free domains is greater when the ice edges are more meridional. Hence,



the integrated OSSC along ice edges that are more meridional is higher than along those that are more zonal. Therefore, $SeaIce\frac{3\pi}{4}$ and $SeaIce\frac{\pi}{4}$ (Fig. 7f,d) can lead to a higher area-integrated OSSC along the ice edge than $SeaIce\frac{5\pi}{8}$ and $SeaIce\frac{3\pi}{8}$ (Fig. 7c,e). Accordingly, the streamfunction responds more sensitively to the percentage of wind stress transferred to the ocean when orientation of the ice edge is more meridional (i.e. diagonal ice edge, denoted by dark pink and dark green lines in Fig. 11).

For all simulations, the simulated gyre transport is always quasi-linearly related to the percentage of stress transferred through the ice to the ocean (Fig. 11). For cyclonic forcing with the increase of the percentage of wind stress felt by the ocean, if the angle between the sea ice edge and wind direction is less than $\frac{\pi}{2}$ (i.e. sea ice in the top-left, denoted by dark and pale pink lines in Fig. 11), the simulated Ψ increases monotonically, i.e., enhances the cyclonic gyre and opposes the anticyclonic gyre. In contrast, if the angle between the sea ice edge and wind direction is greater than $\frac{\pi}{2}$ (i.e. sea ice in the top-right, denoted by dark and pale green lines in Fig. 11), the simulated Ψ decreases monotonically, i.e., enhances the anticyclonic gyre and opposes the cyclonic gyre.

The lines of the simulated streamfunction sometimes have a discontinuity in gradient (blue dots in Fig. 11). Those turning points occur when the model simulates a dipole of two gyres, or a tripole of three gyres, as mentioned in the end of Sect. 3.1. We discuss examples of this in more detail in the following Sect. 4.2.3 and Fig. 12e,f. At the turning points indicated in Fig. 11, although the weaker gyre(s) of the dipole/tripole do follow the quasi-linear relationship, but the dominant gyre (which is captured at blue dots in Fig. 11) does not.

### 4.2.3 Effect of the percentage of $\tau$ transferred to the ocean

Applying the *StrongCyclonic* wind field and $SeaIce\frac{\pi}{4}$ ice coverage, we gradually change the percentage of wind stress transferred to the ocean (%$\tau$) to isolate its influence on the gyre features. The results are shown in Fig. 12.

As in the 0%$\tau$ simulation discussed in Sect. 4.2.1 and 4.2.2, the 20%$\tau$ simulation has negative OSSC over the entire model domain except along the sea ice edge where it has positive OSSC (Fig. 12a). However, the OSS over the ice-covered domain is higher in the 20%$\tau$ simulation (Fig. 12a) than in the 0%$\tau$ simulation (Fig. 11), which reduces the difference in OSS between ice-covered and ice-free regions, resulting in a decrease of the positive OSSC along the sea-ice edge. Therefore, the 20%$\tau$ simulation generates an anticyclonic gyre centred in the model domain (Fig. 12e), in the same location as the 0%$\tau$ simulation (Fig.10b) but slightly weaker. In addition, a second very weak cyclonic gyre is generated in the ice-free domain (southeastern corner in Fig. 12e).

Likewise, for the 40%$\tau$ simulation, the decreased OSS over the ice-covered domain leads to a decreased magnitude of positive OSSC along the sea-ice edge (Fig. 12b). The anticyclonic gyre found in the 20%$\tau$ simulation almost vanishes in the 40%$\tau$ simulation and only a very weak and small anticyclonic gyre is identified in the southwest corner of Fig. 12f. In the ice-free domain, the very weak cyclonic gyre previously found in 20%$\tau$ becomes stronger (southeastern corner in Fig. 12f). A



weaker cyclonic gyre is also formed in the ice-covered domain (northwestern corner in Fig. 12f), generating a tripole. As mentioned above in Sec 4.2.2, this simulation shows an example of one of the turning points in Fig. 11

In the 80%$\tau$ simulation, the OSS in the ice-covered domain is only 20% smaller than that felt in the ice-free domain, making the OSSC along the sea ice edge very weak (Fig. 12c). The anticyclonic gyre found in the previous simulations near the ice edge has now completely vanished, so the other two weak cyclonic gyres identified in Fig. 12g have merged into a single, stronger cyclonic gyre dominating the whole gyre domain (Fig. 12g), similar to the equivalent ice-free simulation (Fig. 8b).

Finally, the 200%$\tau$ simulation has negative OSSC all over the domain (Fig. 12d), forming a very strong cyclonic gyre (Fig. 12h). Despite the asymmetry of the forcing, with the strongest OSSC in the northwest sector and along the diagonal, the gyre is nearly symmetric and centred on the middle of the domain. As discussed previously (Fig. 10 and Fig. 11), between 80%$\tau$ and 200%$\tau$ there is a steady increase in the cyclonic gyre strength.

Overall, changing the percentage of the wind stress transferred to the ocean in this configuration has a dramatic impact
on the gyre. With increasing transfer of wind stress through the ice, the simulated gyre starts from anticyclonic rotation when there is no transfer, develops to dipole, tripole, and finally reverses to cyclonic. This demonstrates that the percentage of wind stress transferred through sea ice to the ocean alone can regulate the gyre strength, and even change the gyre direction.

### 4.2.4 Effect of the angle between the wind and sea ice edge

Now we consider how the gyre responds to changing the angle of the sea ice edge when the wind field is
*StrongCyclonic* and sea ice completely blocks the wind (i.e. 0%$\tau$). In these experiments, the OSSC over the ice-covered domain is zero and the OSSC over the ice-free domain is always negative. The OSSC along the sea ice edge is the same sign as over the ice-free domain (cyclonic) when the angle between wind and sea ice edge is greater than $\frac{\pi}{2}$, but the opposite sign (anticyclonic) when the direction between wind and sea ice edge is less than $\frac{\pi}{2}$. When the angle is exactly $\frac{\pi}{2}$ there is zero OSSC along the sea ice edge, resulting in a cyclonic gyre forced by cyclonic OSSC in the ice-free domain (Fig. 13b). Both $SeaIce\frac{5\pi}{8}$
and $SeaIce\frac{3\pi}{4}$ generate cyclonic gyres (Fig. 13c,d), as the OSSC is cyclonic over whole model domain. For both $SeaIce\frac{3\pi}{8}$ and $SeaIce\frac{\pi}{4}$, the area-integrated OSSC is anticyclonic (Fig. 10), demonstrating that the anticyclonic OSSC along the ice edge dominates relative to the cyclonic OSSC over the ice-covered domain in these simulations. Anticyclonic gyres are thus generated in both $SeaIce\frac{3\pi}{8}$ and $SeaIce\frac{\pi}{4}$ (Fig. 13e,f). As discussed in Sect. 4.4.4, simulations with more "meridional" ice edges (i.e. $SeaIce\frac{3\pi}{4}$ and $SeaIce\frac{\pi}{4}$) have greater impacts on the OSSC, so generate stronger gyres (Fig. 13d,f). Overall, these
experiments demonstrate how the ice edge orientation relative to the wind forcing can determine the gyre strength and direction.



### 4.2.5 Fast ice combined with mobile ice

As described in Sect. 4.2.1, the simulation with wind field and ice conditions similar to those experienced in the Thwaites gyre region in March 2019 (*WeakCyclonic* wind field, $SeaIce \frac{\pi}{4}$ and $0\%\tau$ ice condition, Fig. 9a) generates an anticyclonic gyre (Fig. 9b) similar to the ADCP observations (Fig. 2). Here we explore a very different ice configuration that may also generate an

anticyclonic gyre and is similar to sea ice conditions observed in the Thwaites gyre region in previous seasons (negative streamfunction; lines falling below the 0 Sv horizontal black line in Fig. 11).

Suppose that the southwestern half of the domain is covered in fast ice transferring 0% of $\tau$ to the ocean, and the northeastern half of the domain is covered in mobile sea ice transferring 200% of $\tau$ to the ocean, as shown in Fig. 14a. The steady-state solution for this scenario is an anticyclonic gyre centred near the middle of the model domain (Fig. 14b). Due to

the strong OSSC along the sea ice edge (Fig. 14a), this anticyclonic gyre is stronger (-1.11 Sv) than all gyres simulated from the original model set up (Fig. 11).

This scenario is reminiscent of the sea ice conditions in late January 2019 (Fig. 4a), when fast ice covered the southern part of the Thwaites gyre region while loose sea ice covered the northern part of the Thwaites gyre region. Hence, the sea ice coverage that occurred in January 2019 might generate or facilitate the Thwaites gyre observed in March. To generate gyres,

all that is required is a spatial difference in the amount of wind stress transferred to the ocean, whether that is through fast ice blocking the effect of the wind, or mobile ice enhancing the effect of the wind.

## 5    Discussion

Regional circulation models have often been used to study ocean gyres (e.g. Meneghello et al., 2021; Regan et al., 2020). However, to comprehensively explore the impact of the ice coverage on the gyre formation through its modification of the

stress imparted to the ocean, we need to isolate the individual forcings and conduct a very large number of experiments to examine how they interact. We therefore invoke an idealised model that represents the surface forcing in a simple manner, excluding other features of the real ocean such as baroclinicity, ice-shelf processes and topography. Some biases may occur due to the lack of those mechanisms. For example, the meltwater injection that is thought to facilitate gyres (Mathiot et al., 2017) is not included, which may explain the differences in the magnitude of $\Psi$ between the simulated gyres (0.58 Sv for PIB

and $-0.42$ Sv for Thwaites) and gyre observations (about 1.5 Sv for PIB, Thurnherr et al., 2014, and $-0.27$ Sv for Thwaites). Our argument is not to dispute the effect of meltwater, but rather to highlight the role of wind forcing and the sea ice conditions.

Since our model has much lower computational costs than other regional models, we can run our model hundreds of times to test the gyre response in different wind-ice combinations and apply the results in different polar oceans under varying conditions. All of our simulated gyres reached steady-state within two months, and respond to changed surface conditions on

a similar time scale. We also tested a 1.5-layer reduced gravity model as a comparison for the barotropic case presented here, with all forcings and model design remaining the same. Although the baroclinic model produced gyres with more intensified





surface currents and a slightly longer spin-up time, the gyres in baroclinic and barotropic model cases have the same direction and similar sizes and transports. To explore the sensitivity of the model results to the width of this marginal ice zone, we created two new ice conditions with the smaller gradients (not shown) in surface stress and weaker OSSC over three gridpoints

(3 km) and four gridpoints (4 km). The simulated gyres have almost the same strengths (both about -0.41 Sv) and same shapes as the simulated gyre from simulations without a wider marginal ice zone. This indicates that the width of the marginal ice zone is not important for gyre generation. Hence, our model reproduces the observed gyre direction and response to ice coverage well.

      Meneghello et al. (2018) suggests that the interplay among sea ice, wind and ocean can affect the wind-driven Beaufort

Gyre dynamics through the influence of sea ice on dampening ocean surface currents, the so-called ice-ocean stress governor. The mechanism focuses mainly on mobile ice and the processes occurring under ice cover (e.g. Meneghello et al., 2018; Elvidge et al., 2016). Relatively little attention has been paid to fixed ice and processes near the boundary of ice coverage in the context of wind-driven polar gyres. Mobile ice may drag the ocean and results in a high OSS below sea ice, while fixed ice, including fast ice and ice shelves, will reduce or entirely block OSS below ice. We have considered both the increase and

decrease of OSS, which included scenarios caused by both mobile ice and fixed ice altering OSS. We further discussed the significant effects of the associated OSSC along the ice boundary on gyre formation and gyre features. Our results are especially useful in Antarctic continental shelf seas where climatological winds are often offshore.

      In Antarctic continental shelf seas, gyres near ice shelves contribute significantly to the spreading of glacial meltwater and its associated heat and nutrients (Zheng et al., 2021; Mankoff et al., 2012). Meltwater can impede sea-ice formation, causing

polynyas, or alter the locations and timing of hotspots of marine productivity (Zheng et al., 2021; Mathis et al., 2007; Mankoff et al., 2012). Gyres may transport heat, which is carried by warm water entrained into the gyre, towards ice shelf cavities (Schodlok et al., 2012). The existence of gyres can cause isopycnal displacement. For cyclonic gyres, isopycnals shoal in the gyre centre, as regularly observed in Pine Island Bay (e.g., Thurnherr et al., 2014; Zheng et al., 2021; Dutrieux et al., 2014; Heywood et al., 2016); for anticyclonic gyres, isopycnals deepen in the gyre centre. Below warm-cavity ice shelves, the water

is stratified with a fresh meltwater-rich upper layer and a warm yet salty mCDW lower layer. This isopycnal displacement may allow warmer mCDW to enter the ice cavity and melt ice shelves (S.T Yoon, personal communication). The intrusion of warm water into the base of warm-cavity ice shelve is via the dense lower layer, so called "salt wedge" (Robel, Wilson and Seroussi, 2021). The water mass exchanges due to gyres may impact the stratification in front of ice shelve and hence affect this salt wedge and the related intrusions of warm water to ice shelves. Similarly, gyres formed near the sea ice edge can

circulate ambient water masses. The redistribution of water masses and the heat and freshwater can contain might feedback on sea ice formation to affect the ice formation and regeneration of ice cover in the next winter. By changing both the ocean stratification and the sea ice cover, gyres will affect heat fluxes in the vicinity of ice shelves, which may in turn influence the heat available for basal melting (e.g., St-Laurent, Klinck, and Dinniman, 2015; Webber et al., 2017). Therefore, it is important to understand the conditions that can generate or modify such gyres.



We highlight the importance of wind-ice-ocean interactions, especially the wind stress curl at the ice edge, to polar ocean gyres. These interactions occur at small scales (of the order of tens of km) that will be poorly resolved by global coupled models. Simulating these processes is also dependent on accurate sea ice conditions including the representation of fast ice and polynyas, so further effort is needed to improve these features in such models.

   Due to the importance of gyres for the regional ocean environment, more observations are required to better understand
the relationship between surface conditions and gyre strength and direction. Gyres in typically ice-covered regions (such as Thwaites gyre) present extreme challenges for repeat ship-based surveys. The PIB gyre changes its location interannually and seasonally (Heywood et al., 2016; Zheng et al., 2021), demonstrating the need for long-term continuous monitoring. Such continuous gyre observations would also be useful for model evaluation and could be obtained with high-resolution sea ice motion from satellites, which can clearly show the surface current, or with Ice-Tethered Profilers, which can provide the under-
ice current velocity.

   Despite the important role that gyres play, small wind-driven gyres have received limited attention in the Antarctic continental shelf seas. There are a few other gyres documented in the Antarctic continental shelf seas, such as the Prydz Bay Gyre (Smith et al., 1984), and the cyclonic gyre in front of Filchner Ice Shelf (Foldvik et al., 1985), but none of them are primarily driven by local wind so they are not discussed here. Model representations of ocean currents show the existence of
some small gyres with radius of about 15-30 km, including PIB gyre, in Antarctic continental shelf seas (e.g. Fig. 4c,d in Nakayama et al., 2019). However, except for the PIB gyre, which has been observed in several different years, little attention has been paid to such gyres and their formation mechanisms. This is partly because polar observations often cover either too short a time period or too small an area to provide in situ verification for the gyres found in models. Our study provides a possible mechanism to explain the formation of the gyres formed near ice/ocean boundary that might be explored for other
small gyres shown in high resolution ocean model results in Antarctic continental shelf seas.

   In conclusion, this study uses new observations to identify the Thwaites gyre for the first time, located in a habitually ice-covered region of the southeastern Amundsen Sea. This gyre rotates anticyclonically, despite the climatological cyclonic wind forcing that implies the gyre should rotate cyclonically, as is the case for the only other gyre reported in the eastern Amundsen Sea, PIB gyre (e.g. Thurnherr et al., 2014; Heywood et al., 2016). To investigate this apparent discrepancy, we use a barotropic
model with idealised sea ice and wind forcing only to simulate gyres similar to those observed in the vicinity of ice shelves. Our model suggests that sea ice plays a key role in mediating the wind stress transferred to the ocean and hence determines the direction and strength of the gyre rotation. The percentage of wind stress transferred to the ocean, and the angle between the wind direction and sea ice edge, can alter the OSSC over ice-covered regions and along sea ice edges sufficiently to reverse gyres. Although the simulated gyres are slower than those observed, we demonstrate the potential of sea ice to control gyre
direction and intensity. We suggest that these processes may explain gyre formation and reversal in polar oceans, for example, the PIB gyre reversal hypothesised by Webber et al. (2017). We further suggest that this wind-ice-ocean interaction may contribute to the development of gyre features throughout the polar oceans.



*Code availability.* Model code and code for generating wind forcings used in this study are freely available at
https://github.com/YixiZheng/GyreModel.

*Data availability.* Processed SADCP and CTD data are freely available from the British Oceanographic Data Centre
(www.bodc.ac.uk).

*Author Contribution.* YZ, DPS, KJH and BGMW initiated the concept of this paper and designed the model experiments. BYQ
led the NBP-1902 cruise SADCP measurements collection and identified the Thwaites gyre. YZ ran the model experiments
and visualised the results with help from DPS, KJH and BGMW. YZ analysed the CTD date and calculated the meltwater
content. YZ wrote the manuscript with help from DPS, KJH, BGMW and BYQ. All authors discussed the results and
contributed to the final manuscript.

*Competing interests.* The authors declare that they have no conflict of interest.

*Acknowledgement.* We thank the scientists, technicians and crew working on the NBP-1902 cruise, for everyone's hard work
during the cruise to make the data collection possible. We are grateful to Robert D. Larter (British Antarctic Survey), the chief
scientist of NBP-1902, for his patience and support onboard.

*Financial support.* This work is from the Thwaites-Amundsen Regional Survey and Network (TARSAN) project, a component
of the International Thwaites Glacier Collaboration (ITGC). Support from National Science Foundation and Natural
Environment Research Council (NERC: Grant NE/S006419/1). Logistics provided by NSF-U.S. Antarctic Program and
NERC-British Antarctic Survey. ITGC Contribution No. ITGC-059. This work is also funded by the European Research
Council (H2020-EU.1.1.) under research grant Climate-relevant Ocean Measurements and Processes on the Antarctic
continental Shelf and Slope (COMPASS, grant agreement ID:741120). Y.Z. is supported by China Scholarship Council and
the University of East Anglia.

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



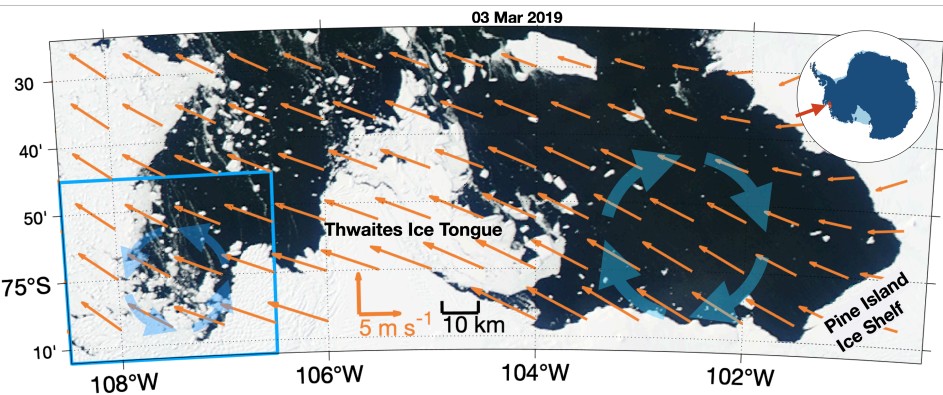


Figure 1. Map of the south-eastern Amundsen Sea. Schematics of Pine Island Bay and Thwaites gyres are shown by thick blue arrows in front of Pine Island Ice Shelf and to the west of Thwaites Ice Tongue, respectively. The ice imagery is from Worldview Aqua/MODIS corrected reflectance (true colour) on 3 Mar 2019. The climatological 10-m wind velocity from 2009-2019 ERA5 reanalysis data (0.25°×0.25° resolution, for clarity, velocity data are interpolated to 0.25×0.125°

resolution) is denoted by orange arrows (see orange scale vectors). The blue box covers the region of Thwaites gyres that is used for Figures 2 and 4. The inset map shows our study region.




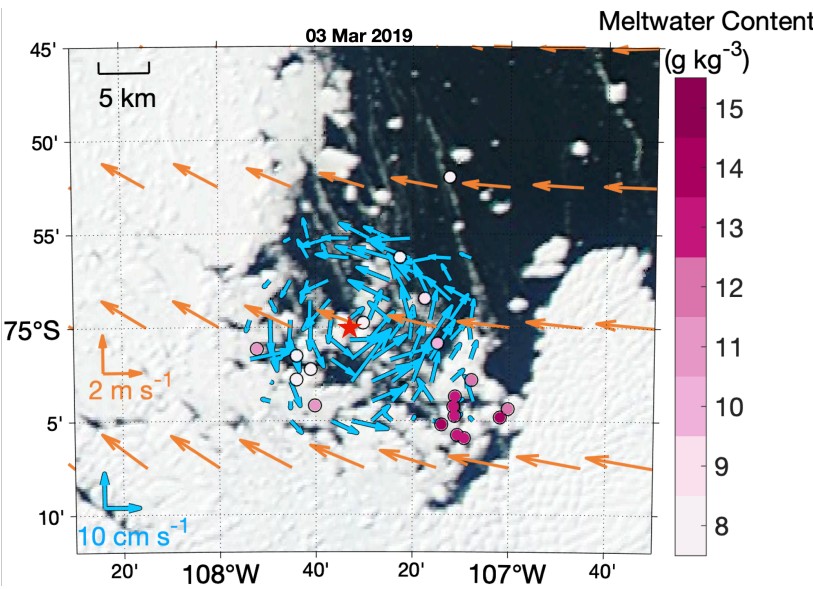

Figure 2. Map of the Thwaites gyre region. The blue arrows indicate the depth-averaged (30-430 m) current velocity from sADCP measurements (observations are averaged to 2km × 2km bins, see blue scale vectors). The red star indicates the gyre centre. The orange arrows indicate the 2009-2019 climatological wind from the ERA5 reanalysis (0.25°×0.25° resolution, for clarity, velocity data is interpolated to 0.25×0.125° resolution, see orange scale vectors). The pink-shaded dots show the full-depth-averaged meltwater content calculated from ship-based CTD data (see colorbar).



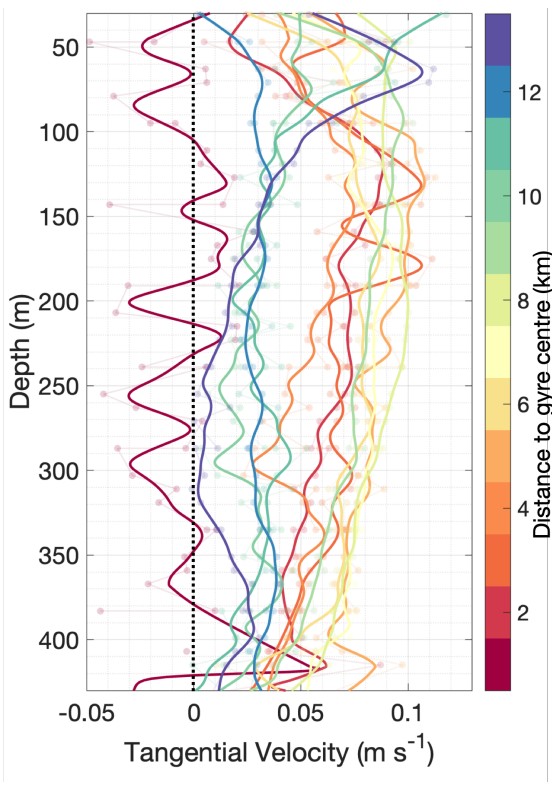

Figure 3. Tangential velocity of the Thwaites gyre with distance to the gyre centre (colours). The binned velocity profiles are denoted in pale dots and lines while the 30-m running-mean smoothed profiles are denoted by thick lines.

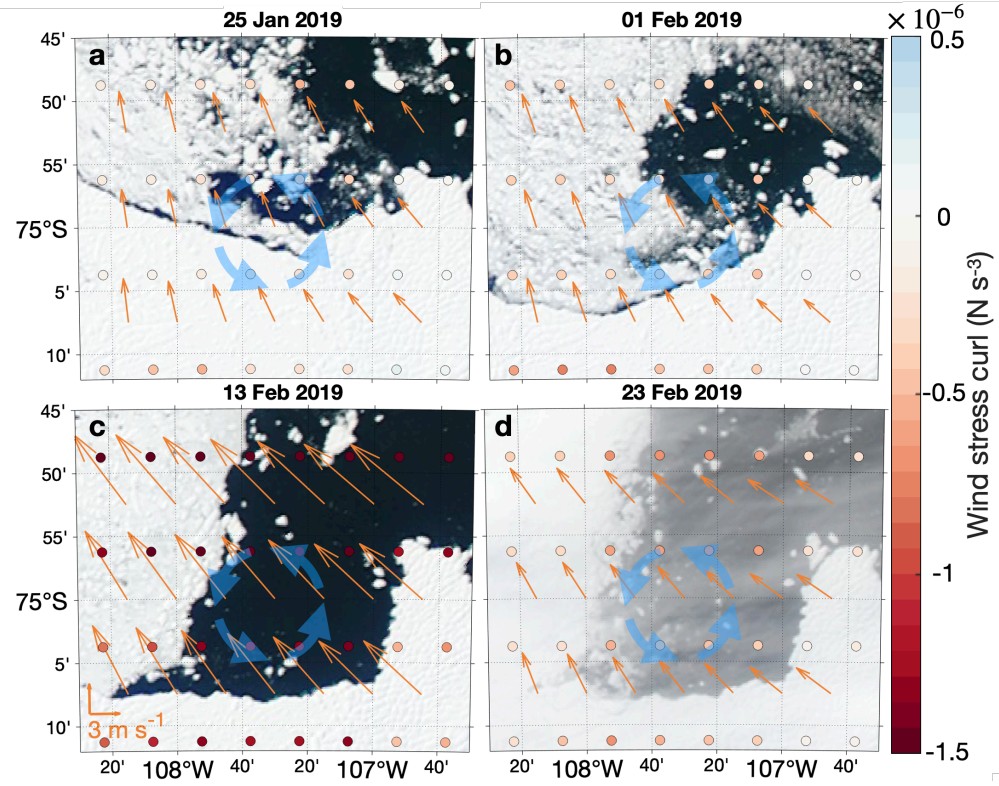

Figure 4. The ice conditions in Thwaites gyre region. The orange arrows denote daily-averaged wind speed from the ERA5 reanalysis (0.25°×0.25° resolution, for clarity, velocity data is meridionally-interpolated to 0.25×0.125° resolution). Dots are coloured by the wind stress curl calculated from the interpolated 0.25×0.125°-resolution daily-averaged wind speed from the ERA5 reanalysis. Thick blue arrows indicate the Thwaites gyre. Ice imagery are from a. 25th Jan, b. 1st Feb, c. 13th Feb and d. 23rd Feb in 2019, respectively, as stated above each panel.





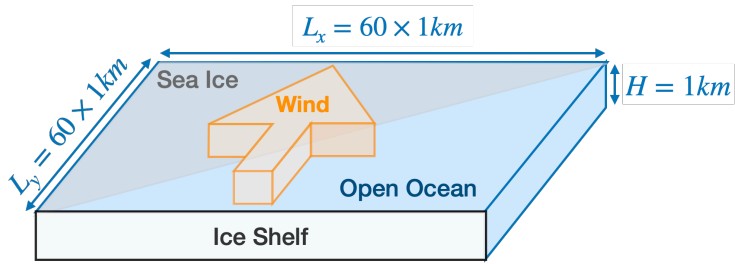

Figure 5. Model schematic. Sea ice is indicated by the pale grey patch covering the northwestern half of the gyre domain. The orange arrow indicates the wind direction, perpendicular to the ice shelf front.






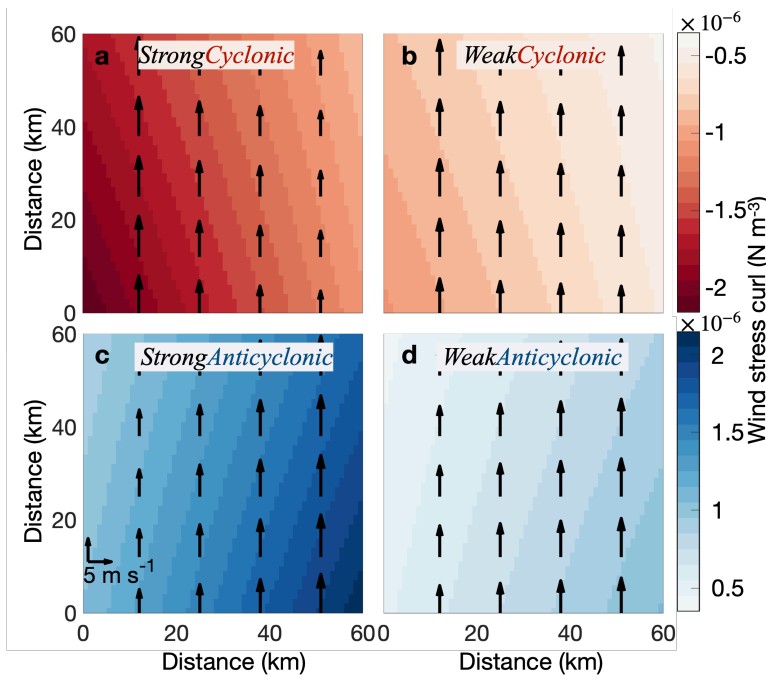

Figure 6. Wind fields applied in this study. a. The wind field representing simplified climatological conditions in the
southeastern Amundsen Sea. b. Same as a, but with wind stress curl reduced by 50%. c. Same as a, but anticyclonic. d. Same
as c, but with wind stress curl reduced by 50%. The arrows show wind stress (only every 13th arrow is plotted for clarity),
with the scale on the southwestern corner of c. Shading shows wind stress curl, red for cyclonic and blue for anticyclonic.



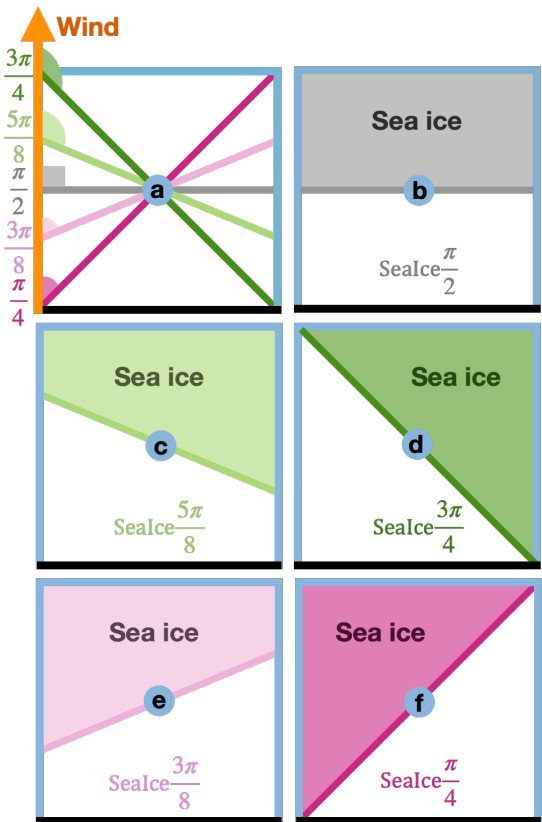

Figure 7. Ice coverages applied in this study. a. Comparison of all ice coverages. Angles are between the ice edges (thick

coloured lines) and wind direction (thick orange line). Thick black line denotes the ice shelf front. b-f. Schematics of sea ice

coverage (shaded patches) and sea ice edge (thick coloured lines).
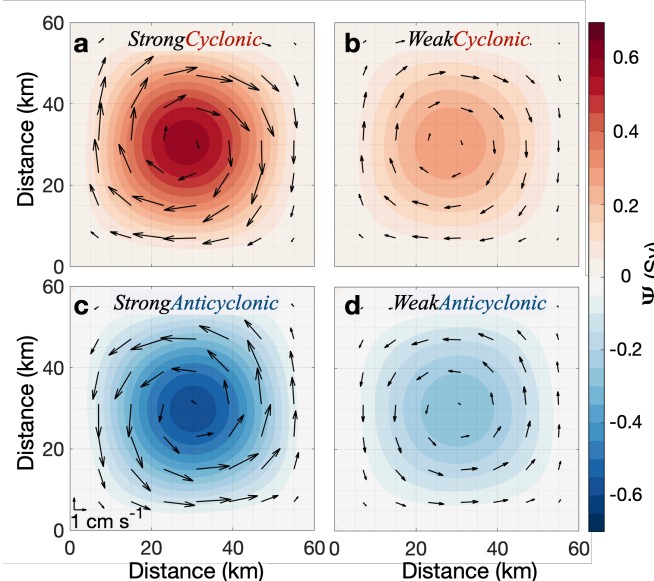

Figure 8. Simulated steady-state gyre stream function (Ψ) and current velocity, when sea ice is absent. Shading indicates streamfunction while arrows indicate current velocity (only every 8th arrow is plotted for clarity). The wind forcing for each of panels (a)-(d) is shown in Figure 6 (a)-(d).



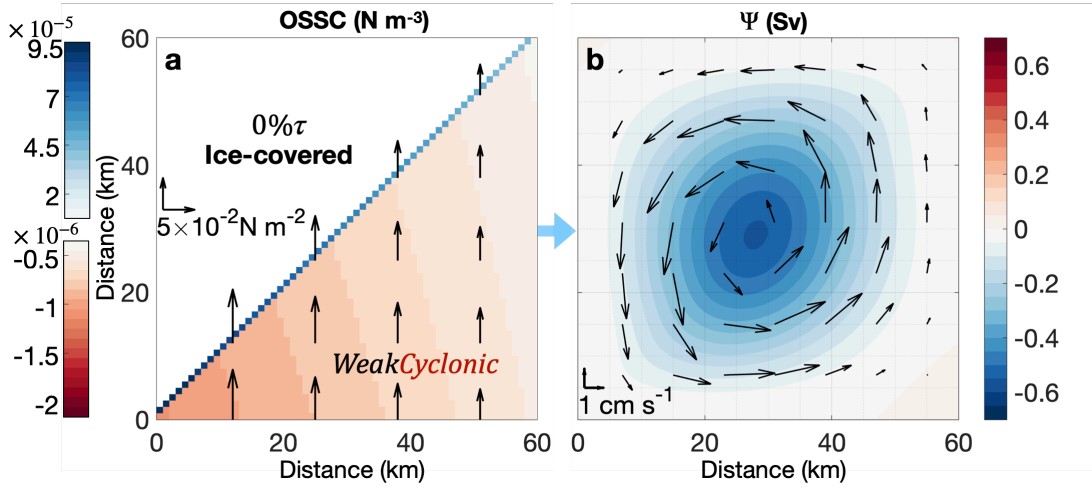

Figure 9. Ocean Surface Stress Curl (OSSC) and simulated gyre streamfunction (Ψ) for simulation with the *WeakCyclonic* wind field and the $SeaIce\frac{\pi}{4}$ and $0\%\tau$ ice coverage. a. Shading indicates OSSC and arrows indicate wind stress (only every 13[th] arrow is plotted for clarity). b. Shading indicates simulated gyre Ψ and arrows indicate current velocity (only every 8[th] arrow is plotted for clarity).







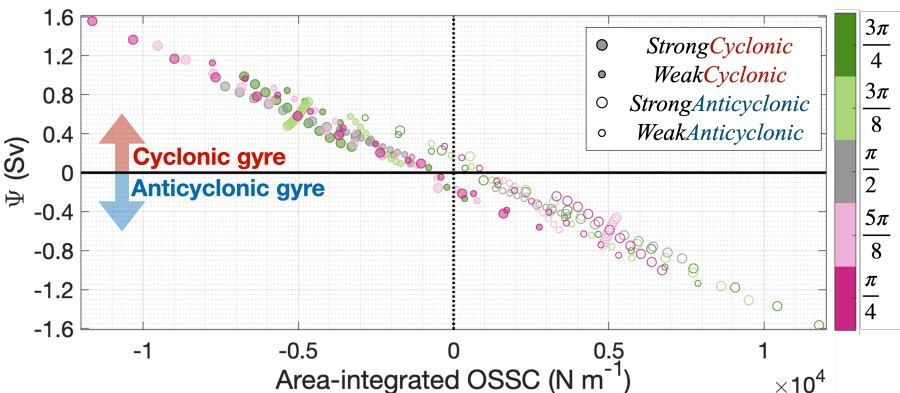

Figure 10. Simulated maximum streamfunction (Ψ) changes with area-integrated OSSC over model domain. There are eleven filled/open circles for each combination of different sizes and colours, indicating simulations with eleven percentages of wind stress transferred to the ocean. Positive streamfunctions are cyclonic (in southern hemisphere) and negative streamfunctions are anticyclonic (in southern hemisphere), while the opposite is true for OSSC. The colours of circles indicate the angle between sea ice edge and ice shelf front, as shown in Fig. 7. Filled circles denote the simulations with cyclonic wind stress curl forcings while open circles denote the simulations with anticyclonic wind stress curl forcings. Large circles denote the simulations with strong wind stress curl forcings while small circles denote the simulations with weak wind stress curl forcings.





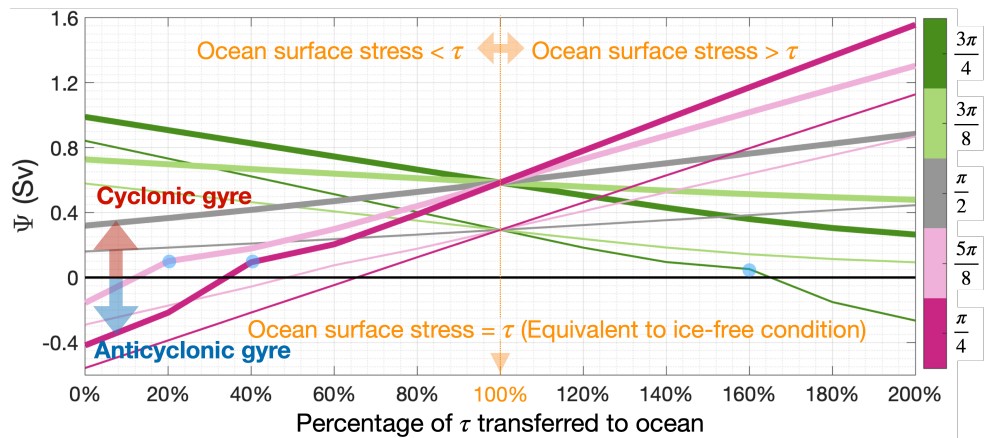

Figure 11. Simulated maximum gyre streamfunction (Ψ) for cyclonic wind stress curl experiments. Positive streamfunction values are cyclonic and negative values are anticyclonic. The line colours indicate the angle between the wind direction and the sea ice edge, as shown in Fig. 7. Thick lines denote the simulations with strong wind stress curl (i.e. *StrongCyclonic*; Fig. 6a) while thin lines denote the simulations with weak wind stress curl (i.e. *WeakCycloic*; Fig. 6b). Blue dots mark the turning points where the lines of simulated Ψ have a discontinuity in gradient caused by the occurrence of dipoles or tripoles.



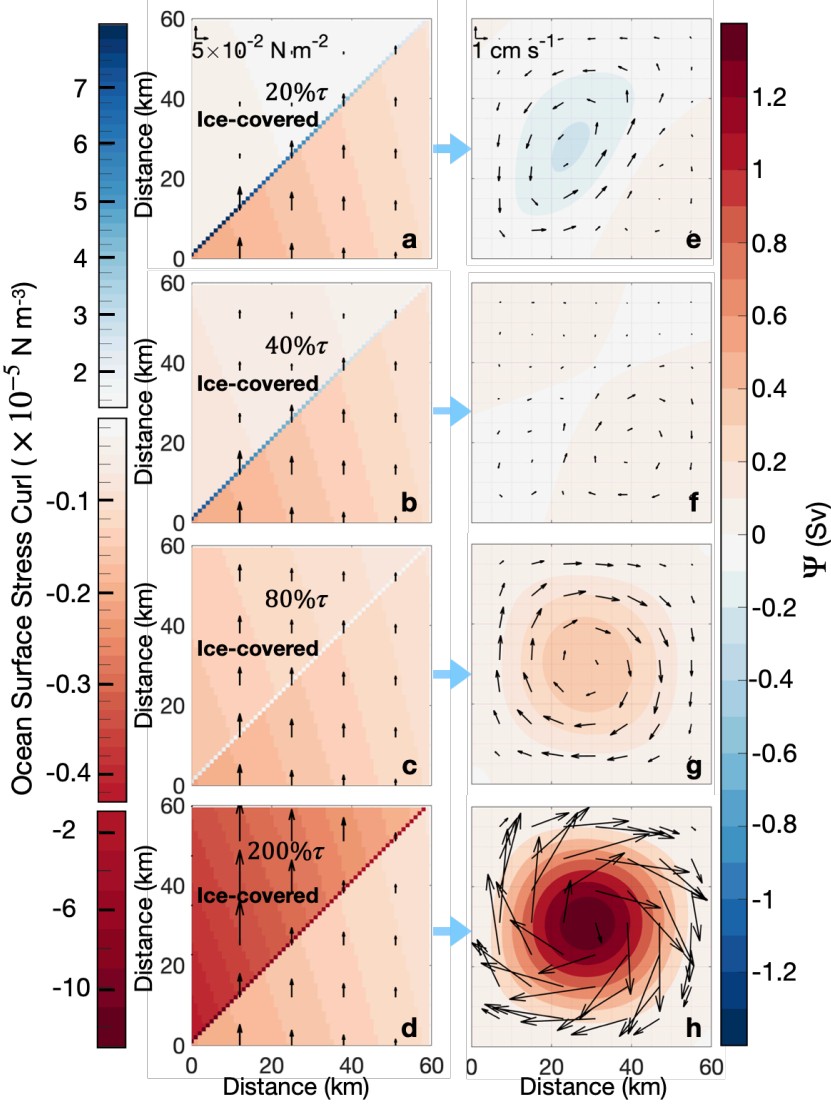

Figure 12. (a-d) Ocean surface stress (arrows, only every 13[th] arrow is plotted for clarity) and ocean surface stress curl (shading) for (a) 20%$\tau$ (b) 40%$\tau$ (c) 80%$\tau$ (d) 200%$\tau$ transferred to the ocean. Sea ice covers the northwestern half of the gyre domain (i.e. *SeaIce* $\frac{\pi}{4}$) and *StrongCyclonic* wind field is applied. (e-h) Simulated streamfunction ($\Psi$, shading) and ocean current velocity (arrows, only every 8[th] arrow is plotted for clarity) resulting from the forcing in panels a-d respectively.





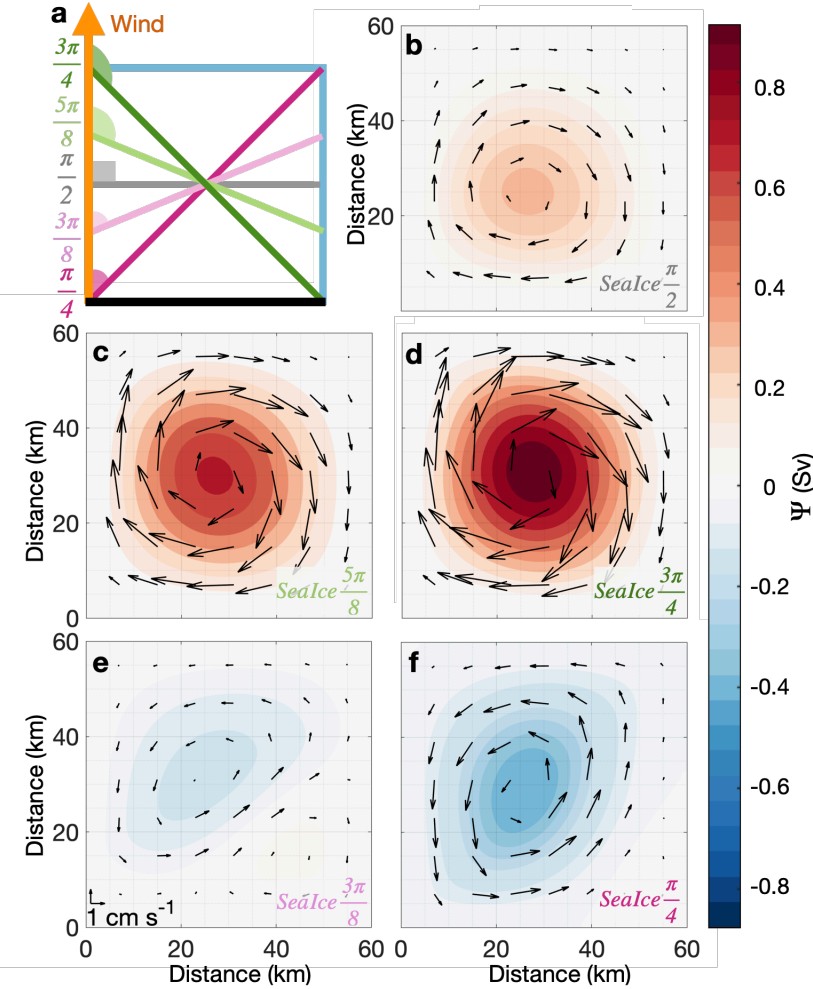

Figure 13. Simulated gyre streamfunction ($\Psi$) when wind field is *StrongCyclonic* and sea ice completely blocks the wind
(i.e. 0%$\tau$). Arrows indicate the current speed (only every 8$^{th}$ arrow is plotted for clarity). Sea ice coverage information for
panels b-f is shown in Figure 7 b-f.



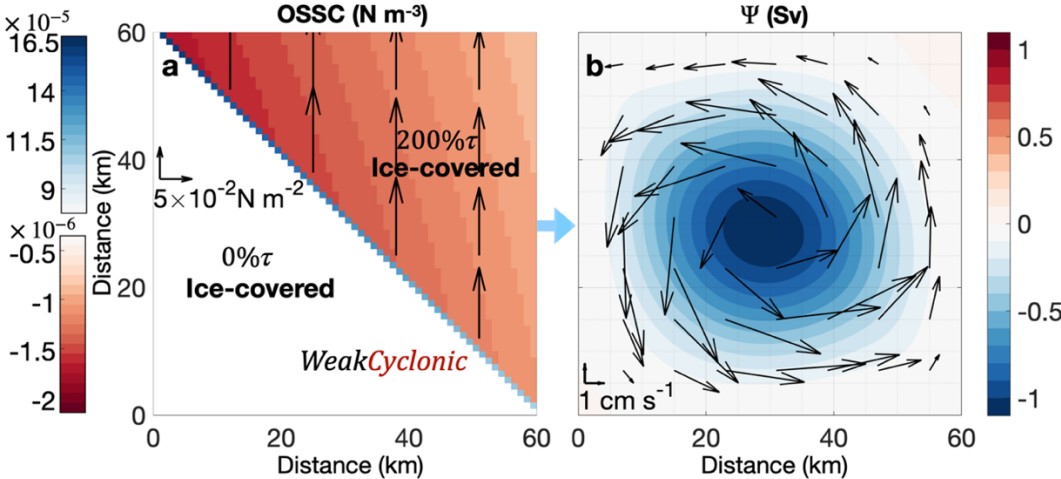

Figure 14. Ocean surface stress curl (OSSC) and simulated gyre streamfunction (Ψ) for simulation with *WeakCyclonic* wind
field. Sea ice covers the whole model domain. Sea ice in the northeast has 200% of $\tau$ transferred to the ocean (representing
mobile ice) while sea ice in the southwest has 0% of $\tau$ transferred to the ocean (representing fast ice). a. The shading
indicates the OSSC and the arrows indicate the wind stress. b. The shading indicates the simulated gyre Ψ and the arrows
indicate the current velocity.