# Peer review of "Reversal of ocean gyres near ice shelves in the Amundsen Sea caused by the interaction of sea ice and wind"

_The Cryosphere, 2021_

## Author Response (AR1)

We thank the reviewers and editor for their constructive comments, which greatly improved our manuscript. We have revised the manuscript according to the reviewers' comments. The detailed response letter has been published in the *Interactive Discussion*, as attached below; and we have made all modifications we mentioned in the *Response Letter*. In addition to the modifications listed in the *Response Letter*, we made some further minor changes

1. We added a line in the *Acknowledgement* as "We thank Shenjie Zhou (British Antarctic Survey) for helping Y.Z with debugging the MITgcm codes".
2. As mentioned in the revised manuscript line 77-80, we made further quality control of our ADCP data. This then affects Figure 2 slightly but the key message is unchanged.
3. We changed the format of the *References* according to the editorial guide.
4. We corrected some typos that we missed in the previous version.

Apart from the formatting, all modifications we made are marked in the track-changes file.

**Response Letter**

Response to the reviewers is in blue, while the modification to the MS is in green.

The authors claim that they identified Thwaites gyre for the first time. They conducted idealized simulations to investigate how sea ice affects gyre strengths and directions, which possibly impact the amount of warm mCDW intruding into ice shelf cavities. This is a nice paper. I do not have many concerns except for one point (see below).

Thank you very much for the positive comments on the manuscript. We will implement the suggested changes as detailed below.

Major comments

(1) If you would like to state that this paper identifies Thwaites gyre, the authors should provide a bit more detail about the observations and structures of the gyre. For example, vertical sections (T, S, Velocity), the ratio between barotropic and baroclinic components, etc.

[Figure]

Revised Fig. 3. Vertical structure of the Thwaites gyre. **a**. Tangential velocity of the Thwaites gyre with distance to the gyre centre (colours). All velocity profiles are horizontally averaged

into 1-km-radius bins (pale dots and lines) then vertically averaged into 30-m bins (thick lines). **b-d.** Section plots of CTD measurements collected in Thwaites gyre region, with distance to the gyre centre. Potential-density isopycnals (in kg m$^{-3}$) are denoted by grey contours. Positions of profiles are marked as triangles at the top of the panel. Below 650m, the water column is occupied by modified Circumpolar Deep Water and is relatively stable. Conservative temperature above freezing is presented in **b**. Absolute Salinity is presented in **c**. Meltwater content is presented in **d**.

Thanks for the good suggestion. We will provide vertical sections across the gyre for T, S and meltwater content in the revised version, as shown above.

Regarding the ratio between the barotropic and baroclinic components, we could not obtain a full-depth velocity measurement to calculate barotropic velocity since the sADCP velocities did not extend down to the sea bed. The vertical current structure can be seen in Revised Fig. 3, and implies that the gyre is a likely to be a combination of baroclinic and barotropic components. We calculate the averaged vertical shear of the tangential velocities between 3—7 km from the gyre centre to quantify this baroclinicity, defined as the vertical gradient of the tangential velocity between the velocity maximum at 130 m, and 620 m, where the tangential velocities are consistent and small. The gyre velocity decreases with depth at a relatively constant rate between these depths. The calculated averaged vertical shear is $2 \times 10^{-4}$ s$^{-1}$, i.e. a change of 0.1 m s$^{-1}$ over 490 m.

Minor comments

Lines 35-40 Authors define PIB gyre. However, I think that Schodlok et al., 2012 do not argue the importance of small gyre circulation in front of PIG. They showed the importance of gyre circulation, which is larger than small gyre circulations discussed in this manuscript, which is confusing to me. A better definition of PIB gyre may be required.

Thank you for pointing it out. We cited Schodlok et al. 2012 because, in its page 160, in the right to the Figure 5b, it says (in orange) "The significant correlation (r= 0.42) between the melt rate and the strength of the Pine Island Bay streamfunction indicates that the **increased basal melt rates of the PIG ice shelf may be due, in part, to local circulation changes**…… On the one hand, there is little change in the **main part of the gyre, which is adjacent to (but does not cross) the shelf break** [*that is the big gyre*], indicating that the flux across the shelf break is not coupled to the continental shelf circulation. On the other hand, **the strength of the smaller gyre, just outside the PIG cavity** [*that is our small PIB gyre*]**, is amplified during the period of high melt rate**, **indicating that circulation on the shelf is more important than the initial provision of CDW across the shelf break**. …… Our study **finds maximum heat transport to the sub-ice-shelf cavity** at the end of summer and beginning of autumn, which is **associated with a maximum in the streamfunction of the small gyre, just outside the PIG cavity** [*that is our small PIB gyre*]" which shows the importance of our small PIB gyre to the melt rate of PIG.

However, we understand why it is confusing as this paper indeed does not emphasise the importance of this small PIB gyre to be their main finding. We will therefore modify this part of text as "Gyres play an important role in local ocean circulation, distributing heat and enhancing water mass exchange in the Amundsen Sea (Zheng et al., 2021; Schodlok et al., 2012). Schodlok et al. (2012) use a high-resolution model to infer that **the strength of this**

**small PIB gyre can be the main determinant** of heat transport toward the ice shelf and the associated glacial melt rate (Schodlok et al., 2012)."

Lines 35-43 It would be nice to see a paragraph on simulated gyre structures (or at least directions) in observations and the existing simulations. For example, you mention Schodlok et al., 2012 show that the strength of the PIB gyre is the main determinant of heat transport. Can you judge from their papers if they simulate gyres in front of PIG? Do previous models show gyres in the same direction or do some models show opposite circulation? As far as I can find, simulated circulation in Nakayama et al., 2014 (Ocean Modelling) is also reversed. Do you find some other examples? Additionally, do you also find something similar to Thwaites gyre in the existing model simulations? If not why? I understand that the authors added a few sentences on this point in the Discussion (Line 351-360) but it would be nice if you could summarize existing model studies with a focus on the representation of eddies in from of Pine Island, Thwaites, etc in the Introduction. Reviews on existing model studies would be helpful.

Thanks for the good suggestion. We will add a few sentences in the Introduction. Regarding the papers the reviewer mentioned: yes, as mentioned above, Schodlok et al., 2012 simulated gyres in front of PIG, and it is cyclonic in their model. We carefully read the Nakayama et al. (2014, Ocean Modelling) paper but we did not find any content about any gyre, eddies or local recirculation - could the reviewer please specify what circulation is reversed? Or perhaps the reviewer is thinking of a different paper? We would welcome any further references that the reviewer is aware of.

All papers we found (based on both models and observations) about the PIB gyre have been mentioned in our MS. We will add a short sentence about a new study focusing on PIB gyre modulating heat flux toward ice shelf (Yoon et al., 2022) in the Introduction. We did not find any other simulations that produce similar features like the Thwaites Gyre. The reasons for that could be: 1. Thwaites gyre region is habitually ice-covered, and 2. gyres with sizes like the Thwaites gyre in model simulations are often overlooked due to the lack of model validation with in-situ observations. We will emphasise the lack of information from models in the Discussion in the revised paper.

Lines 361-362: See major comment above.

We will provide discussion here of the new figures to provide more details of the observed Thwaites gyre.

Figure 7: Sea ice coverage is indicated by shaded patches. It is a bit confusing as initially, I thought white patches are sea ice. Maybe you could state "No sea ice" or something similar to clarify?

Thanks for pointing it out. We will make changes accordingly in the revised version.

Reviewer 2 --------------------------------------------------------------------------------------

**Response Letter**

Response to the reviewers is in blue, while the modification to the MS is in green.

The authors address an intriguing question regarding the ocean circulation in front of Antarctic Ice Shelves. The presented observations in front of the Thwaites Ice Shelf are novel themselves with otherwise little knowledge on the ocean circulation close to the ice shelves. Since the ocean currents deliver oceanic heat to the vulnerable ice shelves in the Amundsen Sea and export melt water away from the ice shelves, exploring the circulation patterns and their drivers are of great relevance. The authors address the question of what determines the direction of the gyres observed in front of the Pine Island and Thwaites Ice Shelves, and they use an idealized barotropic model set up with MITgcm to systematically study the ocean response to different wind stress and sea ice conditions. The model simulations show how the gradients in surface stresses across the sea ice edge superimpose the wind-induced stresses. The strong impact of the sea ice on the gyre strength and circulation highlights the need of good knowledge on sea ice conditions - both in terms of good resolution of sea ice concentration and the impact of the sea ice on ocean surface stresses - to accurately model the ocean circulation.

The results and the methods are well presented and certainly of interest for readers of JPO and the general GFD community. The study is a great combination between novel observations and a systematically approached idealized modeling study, and the figures are clear and easy to understand. I recommend publication of the manuscript and suggest the following changes mainly on the text that are meant to improve the readability, and to emphasize the motivation and the importance of the findings.
Thank you very much for the positive comments on our MS.

General comments on the introduction/ abstract:
The circulation in the PIB is well described (ll 35-49) and the observed reversal of the circulation in PIB by Webber et al (2017) is a great motivation to study the mechanisms that alter the gyre circulation in an idealized framework. From the abstract it is not clear that an anticyclonic circulation has been observed previously.

Thanks for pointing this out. We will add a new sentence to clarify this in the revised version as follows: "A cyclonic (i.e. clockwise) gyre circulation (27 km radius) in front of the Pine Island Ice Shelf has previously been identified in both numerical models and velocity observations. **Mooring data revealed a potential reversal of this gyre during an abnormally cold period**".

While I understand that the PIB gyre is the most observed one (and the observed reversed circulation is a good motivation of this study), the importance of the study could be further highlighted discussing other gyres/ the little attention that has been on small wind-driven gyre. I.e. ll 351-360 could be moved into the introduction. As by now the introduction generally has a strong focus on the PIB circulation, while the circulation in the Thwaites area only follows in section 2. Parts of section 2 could be moved to the introduction.
Thanks for the good suggestion. We will modify and move line 351-360 to the Introduction accordingly. We will also remove part of Sec. 2, regarding how inaccessible Thwaites gyre region is (line 69-70), to Introduction.

ll 53-54 (and 108): The difference in sea ice coverage in the two regions is nicely mentioned as a possible mechanism to alter the gyre circulation. Given the strong focus of the impact of sea ice on ocean circulation in your study, it would be nice to expand a little on how sea ice coverage influences ocean surface stresses, i.e. having one paragraph describing what determines the local circulation and the importance of including sea ice in the calculation of ocean surface stresses. Only later in ll. 144-149 it is mentioned that sea ice can both increase and reduced the ocean surface stresses, but it should be mentioned already in the introduction (ll. 314-322 could be moved to introduction).

Thanks for the good suggestion (your suggestions indeed make our Introduction much more comprehensive so that non-oceanographer readers can understand everything more easily!). We will expand ll. 314-322 to an extra paragraph discussing how previous work reveals that the sea ice can alter ocean surface stresses and move them to the Introduction.

Specific comments:
ll 18: I suggest writing something like "a range of idealized sea ice conditions typical for the region" without mentioning the satellite images, given the very idealized setting used and the large range of different angles and stress transfers explored in this study.

Thanks for pointing it out. We will change "simplified sea ice conditions from MODIS satellite images" to "a range of idealised sea ice conditions typical for the region" in our Abstract. The ice conditions we used that reproduced the PIB and Thwaites gyres are, however, designed based on satellite images. Those experiments are our "case study" that we use to show that our model can be applied to real gyres, so sentences describing how they were designed will be retained.

ll 51: Highlight here that a cyclonic wind field would induce a cyclonic gyre circulation in the open ocean without any other influences (only mentioned in ll. 75-76)

Thanks for the good suggestion. We will change this sentence to "Influenced by the same climatologically-cyclonic wind field (which favours cyclonic gyres), Thwaites gyre is anticyclonic, raising the intriguing question of what mechanism(s) control(s) the direction of these gyres" in the revised version.

ll 73 and Figure 3: Any reasons why the velocities are only shown in the upper 450m when the currents were measured in the upper 980m? Please elaborate.

[Figure]

Revised Fig. 3. Vertical structure of the Thwaites gyre. **a**. Tangential velocity of the Thwaites gyre with distance to the gyre centre (colours). All velocity profiles are horizontally averaged into 1-km-radius bins (pale dots and lines) then vertically averaged into 30-m bins (thick lines). **b-d.** Section plots of CTD measurements collected in Thwaites gyre region, with distance to the gyre centre. Potential-density isopycnals (in kg m$^{-3}$) are denoted by grey contours. Positions of profiles are marked as triangles at the top of the panel. Below 650m, the water column is occupied by modified Circumpolar Deep Water and is relatively stable. Conservative temperature above freezing is presented in **b**. Absolute Salinity is presented in **c**. Meltwater content is presented in **d**.

Thanks for pointing it out. We cut the profiles at 430 m because below that the Thwaites gyre velocities are weak. We produced a revised figure using the full depth range of good velocity data, accompanied by transect plots from CTD data for a detailed structure of the Thwaites gyre (shown above) and will include it in the revised version.

ll 144: It was not clear to me at first how sea ice is represented in the model. "No active sea ice model" can be understood as if there was a sea ice model included that simply does not interact with the ocean thermodynamically. Please clarify that no sea ice is included in the model simulations, but accounted for by changing the strength of the ocean surface stresses in the sea ice covered area.

Thanks for the good suggestion. We will modify this sentence to "We do not include a sea ice model in our study, but change the strength of OSS to simulate the influence of sea ice in the ice-covered area" in the revised version.

ll 157: What do you mean by "unchanged"?

Thanks for pointing it out. To reduce confusion, we will change the related sentences from "When sea ice is absent, the OSS and OSSC remain unchanged over the whole model domain. The strength and direction of gyres solely depend on the wind field" to "We first consider results from simulations with no sea ice coverage. Here the strength and direction of gyres solely depend on the wind field" in the revised version.

ll 173-175: The OSS reduction by sea ice (0% tau) is not mentioned in the text (only in Figure 9a).

Thanks for pointing it out. We will modify it to "We therefore consider the *WeakCyclonic* wind field and the *SeaIce*$\frac{\pi}{4}$ and **0% τ** coverage" for clarification in the revised version.

ll 187-188: Please quantify the speed and stream function for the 400m depth simulations.

Thanks for pointing it out. The speed and streamfunction for 400m-simulation are 0.58 Sv, 7.6 cm/s, when sea ice is not present and 7.1 cm/s, -0.55 Sv for *SeaIce*$\frac{\pi}{4}$, 0% τ and *WeakCyclonic* (similar to Thwaites gyre). We will add this information to the revised version.

Section 4.2.2: Figures 10 and 11 neatly summarize the impact of the ocean surface stress and the sea ice angle on the ocean circulation. Since both figures contain a lot of information, the section could benefit from an introduction into the figures and a more logic order. For example in line 209, it could be introduced that the figure is for the cyclonic wind stress only and it might be more logic to start with paragraph ll. 222-228.

Thank you for the good suggestion. We will modify the paragraph accordingly, and move ll. 222-228 forwards.

Figure 11 (also in 10): It would be nice to mark the experiments presented in Fig. 8 and 9 for reference.

Thank you for the good suggestion. We will modify the figure accordingly in the revised version.

ll 202-208: I would highlight already here that you find this relationship for most model simulations, but that there are certain conditions under which the circulation can be reversed. This makes a nice transition to Figure 11 which shows more details on the conditions for reversed circulation.

Thank you for the good suggestion. We will modify the paragraph accordingly

ll 214-221: Fig. 11 is one of the substantial figures in the paper and the logic of the results seems disrupted starting this paragraph with Fig. 10. I suggest framing this paragraph around the results shown in Fig. 11 rather than basing it on Fig. 10.

Thank you for pointing it out. We will swap Fig. 10 and Fig. 11 in the revised version to form a more logical structure.

ll 235: Results are already shown in Fig. 10 and 11. -> The spacial distribution of the OSSC and the circulation pattern are shown in Fig. 12.

Thank you for pointing it out. We will change the sentence to "The spatial distribution of the OSSC and the circulation pattern are shown in Fig. 12" as suggested.

ll 259: specify that it's the wind stress transferred to the ocean by sea ice/ within the sea ice covered area.
Thank you for pointing it out. We will modify it to "Overall, changing the percentage of the wind stress transferred **through ice** to the ocean in this configuration has a dramatic impact on the gyre."

ll. 340-343: Due to the large influence of the sea ice on ocean surface stresses explored in this study, it would be worth to include a broader discussion on the quality of sea ice satellite products, their resolution and the data coverage, as well as the state of the art in terms of sea ice contribution to ocean surface stress. As you showed nicely in your paper, there is a strong need in resolving whether sea ice enhances or reduces the surface stress on the ocean, while the available sea ice products are still far from resolving this information in the needed resolution, as detailed information on sea ice thickness and roughness would be needed.
Thank you for the good suggestion. We will add a few sentences here discussing sea ice satellite products.

ll. 361: I suggest to add a new section: 6 Conclusions, for easier navigation.
Thank you for pointing it out. We will add a Conclusion session in the end of the revised version.

Technical corrections:
ll 51: Influenced by the same climatologically-cyclonic wind field, -> same wind field as what?
Thank you for pointing it out. We'll modify it as "both influenced by climatologically-cyclonic wind field"

ll 69: covered by sea ice all year round?
Thank you for pointing it out. Yes, it's covered by sea ice year-round. We will modify this sentence to "This region is habitually covered by sea ice year-round and has only opened twice since 2000" for clarity.

ll 74: units not in italic (106 m3 s-1)
Thank you for pointing it out. We will modify it to italic format.

ll164: due to the lack OF surface intensification ...
Thank you for pointing it out. We will add the "of".

ll 232: remove 'although'
Thank you for pointing it out. We will remove the "although".

ll. 273: Section 4.4.4 -> 4.2.2
(oops) Thank you for pointing it out. We will change the section number.

ll. 335: remove 'can contain'?
Thank you for pointing it out. We will remove it.

Figure 1: Is it correct that the plotted velocity data are on a 0.25x0.125 resolution? It looks more like 0.5x0.125 in this Figure.
Thank you for pointing it out. The velocity data were interpolated to 0.25x0.125 resolution, but only every other arrow in zonal direction is plotted. We will add a short description to the figure

caption "0.25º×0.25º resolution, velocity data are interpolated to 0.25º×0.125º resolution. Only every other arrow in the zonal direction is plotted for clarity)" in the revised version.